# Exploring the Role of PD-1 in the Autoimmune Response: Insights into Its Implication in Systemic Lupus Erythematosus

**DOI:** 10.3390/ijms25147726

**Published:** 2024-07-15

**Authors:** Nefertari Sagrero-Fabela, Ramón Chávez-Mireles, Diana Celeste Salazar-Camarena, Claudia Azucena Palafox-Sánchez

**Affiliations:** 1Doctorado en Ciencias Biomédicas (DCB), Centro Universitario de Ciencias de la Salud, Universidad de Guadalajara, Guadalajara 44340, Mexico; nefertarisagrero@gmail.com (N.S.-F.); ramon.chavez9890@alumnos.udg.mx (R.C.-M.); 2Grupo de Inmunología Molecular, Centro Universitario de Ciencias de la Salud, Universidad de Guadalajara, Guadalajara 44340, Mexico; celeste.salazar@academicos.udg.mx; 3Instituto de Investigación en Ciencias Biomédicas (IICB), Centro Universitario de Ciencias de la Salud, Universidad de Guadalajara, Guadalajara 44340, Mexico

**Keywords:** PD-1, PD-L1/PD-L2, systemic lupus erythematosus, T follicular helper (Tfh) cells, T peripheral helper (Tph) cells

## Abstract

Despite advances in understanding systemic lupus erythematosus (SLE), many challenges remain in unraveling the precise mechanisms behind the disease’s development and progression. Recent evidence has questioned the role of programmed cell death protein 1 (PD-1) in suppressing autoreactive CD4^+^ T cells during autoimmune responses. Research has investigated the potential impacts of PD-1 on various CD4^+^ T-cell subpopulations, including T follicular helper (Tfh) cells, circulating Tfh (cTfh) cells, and T peripheral helper (Tph) cells, all of which exhibit substantial PD-1 expression and are closely related to several autoimmune disorders, including SLE. This review highlights the complex role of PD-1 in autoimmunity and emphasizes the imperative for further research to elucidate its functions during autoreactive T-cell responses. Additionally, we address the potential of PD-1 and its ligands as possible therapeutic targets in SLE.

## 1. Introduction

Autoimmune diseases are reported to affect 5–8% of the world’s population [1]. They have garnered significant attention in research due to the multitude of immunopathological pathways characterized by aberrant innate and adaptive responses and the absence of effective therapeutic interventions.

Systemic lupus erythematosus (SLE) is considered the prototype of autoimmune diseases due to the complexity of the molecular and immunological pathways involved in its development and progression [2]. Although the production of autoantibodies targeting nuclear antigens is characteristic of this disease, this process involves a wide range of mechanisms. One of the most important is the differentiation and maturation of T-cell-dependent B cells. During this process, CD4^+^ T cells that have evaded central and peripheral tolerance mechanisms collaborate with B cells, providing them with the necessary signals for their complete differentiation into antibody-producing plasma cells or long-lived memory cells [3].

During this interaction, T cells also require co-stimulatory signals for optimal activation, differentiation, and function. However, ultimately, they also require inhibitory signals to maintain immune balance, limit autoreactivity, and prevent tissue injury. These signals, known as immune checkpoints, are added to co-stimulatory signals to inhibit T-cell receptor (TCR) pathway signaling, thus counteracting cellular overactivation and contributing to immune response control. Nevertheless, it has been reported that dysregulation of the function or expression of these checkpoints leads to failure to control T-cell exhaustion, favoring autoimmunity [4].

One of the most important checkpoints is the programmed cell death protein 1 (PD-1) receptor, which has been reported to prevent autoimmunity since its deficiency or blockade of the PD-1 signaling pathway exacerbates disease progression in several autoimmune mouse models [5,6,7,8,9]. In addition, mutations in the *PDCD1* gene are associated with susceptibility to several human autoimmune disorders [10,11,12,13]. PD-1 controls central and peripheral self-tolerance and prevents autoimmunity by suppressing autoreactive T cells. However, it has been reported that effector PD-1^+/high^CD4^+^ T cells are increased in patients with several autoimmune diseases associated with disease activity (reviewed in [14]), like SLE. This suggests that these cells are not functionally restricted by their high expression of PD-1.

This review provides a comprehensive overview of our current understanding of PD-1 and its association with the mechanisms involved in autoimmune responses with a special focus on SLE. Our goal is to provide possible explanations for why PD-1 may not effectively perform its suppressive functions, even when overexpressed on CD4^+^ T cells, and raise important questions about PD-1’s function for future research in this area.

## 2. PD-1 Expression and Function: What Do We Know?

Our understanding of the PD-1 molecule began with identifying its cDNA in 1991 [15], followed by the discovery that PD-1 transcript expression occurred in T cells just before their activation and that its function was associated with the induction of cell death. Interestingly, it was also observed that mice lacking PD-1 expression exhibited abnormal activation of both B and T cells [16]. It was further demonstrated that these PD-1-deficient mice developed diseases, such as cardiomyopathy, arthritis, and nephritis, highlighting its importance not only as a negative regulator of the immune response but also in the development of autoimmune-related diseases [9,17,18].

PD-1 (CD279) is a signal-transducing type I protein composed of 260 amino acids that is expressed on the surfaces of activated T cells, B cells, thymocytes, natural killer (NK) cells, natural killer T (NKT) cells, macrophages, and dendritic cells (DCs) [19,20,21]. Structurally, PD-1 is composed of an extracellular immunoglobulin variable region (called an IgV-like domain), followed by a constant region (called an IgC-like domain), a transmembrane domain, and an intracellular domain. The IgV-like domain of PD-1 contains the binding site for its ligands, while the intracellular domain contains two tyrosine residues: an immunoreceptor tyrosine-based inhibitory motif (ITIM) and an immunoreceptor tyrosine-based switch motif (ITSM) [22,23]. PD-1 has two major ligands, known as PD-L1 (CD274) [24] and PD-L2 (CD273) [25], that share their extracellular structure with PD-1 (Figure 1). PD-1 belongs to the CD28/CTLA-4 superfamily and has 21–33% sequence identity with other family members like CTLA-4, CD28, and ICOS [26].

The PD-1 pathway plays a pivotal role in maintaining immune balance, serving as a negative regulator of T-cell receptor (TCR) signaling and suppressing T-cell activation triggered by auto-antigens. Thus, PD-1 prevents excessive activation following an adaptive immune response [27]. It has also been shown that PD-1 negatively regulates B-cell responses by inhibiting B-cell receptor (BCR) signaling [28]. 

During chronic infections, PD-1 signaling plays a central role in perpetuating T-cell exhaustion, a state in which T cells progressively lose their effector functions after their activation [19,29,30,31]. Furthermore, it has been well documented that tumor cells exploit the PD-1 signaling pathway to evade immune responses. Indeed, antibodies that blockade the activation of this pathway have been widely used in cancer immunotherapies [32]. However, despite extensive knowledge regarding PD-1’s involvement in cancer responses, its role in autoimmunity, particularly in the pathogenesis of diseases like SLE, is still largely unexplored.

## 3. PD-1/PD-L1/PD-L2 in Central and Peripheral Tolerance

The most widely accepted role for PD-1 in autoimmunity is during the mechanisms of autoreactive T-cell deletion during central and peripheral cell tolerance [33]. The initial observations regarding the significance of PD-1 in thymic selection mechanisms were made by Hiroyuki Nishimura et al. in 2000. They investigated the impact of PD-1 deficiency on thymocyte differentiation, finding that PD-1 negatively regulates beta selection while modulating positive selection. Thus, PD-1 significantly alters the mature T-cell repertoire [34]. These findings were subsequently corroborated by others, who also indicated a role for PD-1 during thymic selection [35,36]. More recently, it was reported that mice lacking the expression of both PD-1 and the autoimmune regulator (AIRE) genes developed fatal adulthood autoimmunity. This phenomenon was not observed in mice lacking other combinations of tolerance mediators, suggesting that cooperation between PD-1 and AIRE plays a pivotal role in mediating central tolerance and autoimmune development [37].

The peripheral tolerance of T cells in the context of PD-1 is mediated by the expression of PD-L1 and PD-L2 on antigen-presenting cells (APCs), such as dendritic cells, which determine T-cell inhibition through this expression [38]. However, a decrease in the expression of PD-1 or its ligands, which activate the inhibitory pathway, could be closely linked to the development of autoimmune diseases including SLE (as discussed below). This is due to the lack of inhibition of autoreactive T cells.

In line with this, recent research has focused on two subpopulations of T helper (Th), cells whose frequencies appear to be associated with the pathogenesis of some autoimmune conditions. T follicular helper (Tfh) cells are usually identified by a CD4^+^CXCR5^+^PD-1^+^ phenotype. These cells represent a specialized subpopulation, primarily located within the germinal centers (GCs) of secondary lymphoid organs, which play a crucial role in mediating the differentiation of B cells into autoantibody-producing cells. This function is primarily achieved through the expression of IL-21 [39], which is a key regulator of various processes within the immune system and autoimmunity. It is the main cytokine required to enable germinal center formation, affinity maturation, and the generation of plasma cells and memory B cells [40]. Furthermore, IL-21 is also required for the complete generation of Tfh cells [41,42]. A small proportion of Tfh cells called circulating Tfh (cTfh) cells can be found in the peripheral blood, and they have been extensively associated with autoimmune diseases, such as systemic lupus erythematosus (SLE) [43,44,45,46,47], RA [48,49,50], Sjögren’s syndrome (pSS) [51,52,53,54,55], and multiple sclerosis (MS) [56,57]. Although Tfh cells are the main CD4^+^ T-cell subpopulation involved in the generation of autoantibodies, recent research has focused on a novel subpopulation that appears to be more closely associated with the pathogenesis of autoimmune diseases. 

T peripheral helper (Tph) cells are characterized by a CD4^+^CXCR5^−^PD-1^+/high^ phenotype. The absence of CXCR5 leads them to exit germinal centers and position themselves within inflamed tissues by expressing other chemokine receptors such as CCR2 and CCR5. In these tissues, Tph cells induce B-cell differentiation through the production of IL-21 [14]. Increased numbers of Tph cells have also been observed in SLE [58,59,60], RA [61,62,63,64], and pSS [53,65,66].

A characteristic shared by Tfh, cTfh, and Tph cells is the high expression of PD-1 on their surfaces, which, in some cases, is associated with disease activity. Because the frequencies of cTfh and Tph cells are increased in these autoimmune diseases, it is reasonably suggested that PD-1 may not be exerting its suppressive functions on these cellular subpopulations. This may indicate a break in peripheral tolerance due to an aberrant function of PD-1.

Another important subpopulation of T cells responsible for mediating peripheral tolerance and tissue damage is regulatory T cells (Treg), which are identified by their CD4^+^CD25^+^Foxp3^+^ phenotype. In addition, they express activation markers and immunomodulatory molecules such as CTLA-4, PD-1, and PD-L1 [67]. Treg cells are divided into two groups: those that express Foxp3 and differentiate in the thymus, which are known as natural Tregs (nTregs), and those that differentiate from naïve CD4^+^ T cells in the periphery or through in vitro stimulation, which are known as peripheral (pTreg) and induced (iTreg) Treg cells, respectively [68]. Increasing evidence has shown that the PD-1 pathway is associated with the differentiation of pTreg cells. Francisco, Loise M et al. demonstrated that PD-L1-deficient antigen-presenting cells failed to induce the polarization of naïve CD4^+^ T cells into pTreg cells. When CD4^+^ T cells were primed with PD-L1 in vitro, iTregs were induced, demonstrating that PD-L1 maintained Foxp3 expression and the suppressive function of iTreg cells [69]. This could contribute to our understanding of how a deficiency in PD-L1 expression by APCs is associated with the development of autoimmune diseases. Similar observations have been made for Th1 cells, which, in a PD-L1-dependent manner, polarized to Treg cells [70].

Recently, a higher expression of PD-1 and lower expression of PD-L1 were demonstrated on CD4^+^CD25^+^FOXP3^+^ Treg cells after stimulation in a cell culture. This was negatively associated with the SLEDAI index in SLE patients [71]. Furthermore, it has been shown that the stimulation of PD-1 with specific agonists can inhibit autoreactive T cells and restore Treg cell homeostasis [72], highlighting the importance of the PD-1 pathway for Treg cells and peripheral tolerance (Figure 2).

## 4. Regulation of PD-1 Expression and Its Relationship with the Autoimmune Response in SLE

To understand the role of PD-1 in autoimmune responses in SLE and other autoimmune disorders, it is important to consider how PD-1, its ligands, and the downstream signaling pathways are regulated.

### 4.1. Stimulation of PD-1 Expression

*PDCD1*, which is located on chromosome 2, is the gene encoding PD-1 in humans, whereas *Pdcd1*, which is located on chromosome 1, is its counterpart in mice. PD-1 transcription is initiated by TCR activation and cytokine stimulation, which activate various transcription factors, including NFATc1 (activated by TCR signaling) [73]; STAT3, STAT4, and STAT5 (activated by IL-6, IL-12, IL-2, and IL-21) [74,75]; the ISGF3 complex (activated by IFNα) [76]; and FOXO1, AP-1, and NF-κB (activated by various other cytokines) [77,78]. These transcription factors bind directly to specific enhancers and promoters located in cis-regulatory elements, comprising two conserved regions (Conserved Region B (CR-B) and Conserved Region C (CR-C)) of the PD-1 gene in both humans and mice [79] (Figure 3A).

This could offer an initial explanation for the overexpression of PD-1 on the surfaces of CD4^+^ T cells, like cTfh and Tph cells, especially in autoimmune diseases such as SLE, where there is constant and repetitive stimulation of T cells by self-antigens and proinflammatory cytokines like IL-6, IL-21, and IFNα [2]. After translation, PD-1 is stabilized via fucosylation in the endoplasmic reticulum and Golgi apparatus [80]. The presence of PD-1 has been identified in vesicles proximal to the Golgi and trans-Golgi network, hinting at a potential role for these vesicles as reservoirs for PD-1 awaiting activation through TCRs [81].

### 4.2. Repression of PD-1 Expression

Other transcription factor binding sites near CR-B and CR-C repress PD-1 expression. These include an RBP-Jκ binding site upstream of CR-C [82], a Blimp-1 binding site downstream of CR-C [83], and a T-bet binding site −0.5 kb from *Pdcd1* [84].

Regarding this, it was demonstrated that increased expression of the transcription factor Blimp-1 was correlated with overexpression of PD-1 in CD8^+^ T cells during chronic viral infections. In the same work, it was found that upon the conditional deletion of Blimp-1, there was also a reduction in the expression of this inhibitory receptor [85]. However, a couple of years later, it was demonstrated that PD-1 can be repressed by Blimp-1 in CD8^+^ T cells by suppressing the expression of NFATc1, which acts as a transcriptional activator of *Pdcd1* during acute viral infections [83]. Despite these contradictory results in response to the infection state, it is important to highlight that there is an association between Blimp-1 and PD-1 expression in CD8^+^ T cells. While Blimp-1 could play different roles in CD8^+^ and CD4^+^ T cells, the possible association between this transcription factor and PD-1 regulation in CD4^+^ T cells has not been explored during normal and autoimmune responses, even though Blimp-1 has been linked to exhausted protein markers such as PD-1 [86,87]. Tph cells are characterized by the expression of the Blimp-1 transcription factor, which serves as a mediator of its differentiation. As Tph cells overexpress PD-1 on their surfaces, it could be interesting to study the mechanisms involved in the regulation of the *PDCD1* gene by Blimp-1. 

## 5. PD-1 Ligands Emerged as a New and Important Focus of Study in the Field of SLE

PD-L1 is coded by the *Cd274* gene, which is situated on chromosome 9 in humans and chromosome 19 in mice. Similarly, PD-L2 is encoded by the *Pdcd1lg2* gene, which is found on chromosome 9 in humans and chromosome 19 in mice [88]. PD-L1 and PD-L2 belong to the B7 superfamily. Both are type I transmembrane glycoproteins composed of IgV- and IgC-like domains. Despite PD-L1 and PD-L2 having 30–40% amino acid homology in humans, it has been reported that PD-L1 is the main ligand though which PD-1 activates its downstream pathway [89].

PD-L1 and PD-L2 have distinct expression patterns in various components of the immune system as well as in cells of other tissues [90]. Both are expressed in activated T cells, B cells, dendritic cells, macrophages, and bone marrow-derived mast cells [91]. Additionally, PD-L1 is also expressed by NK cells and non-immunological cells, like endothelial cells, epithelial cells, and keratinocytes [92,93].

PD-1/PD-L1/2 ligation is not restrictive, as PD-L1 can also bind to CD80, while PD-L2 can also bind to repulsive guidance molecule b (RGMb) [94,95,96]. The binding between PD-L1 and CD80 occurs in cis (on the same cell surface) [97,98]. As the activation of the pathway requires the binding of PD-1 with either PD-L1 or PD-L2, the availability of these ligands is necessary for the inhibitory function of PD-1 in T cells. Therefore, PD-L1/CD80 cis binding on antigen-presenting cells (APCs) can reduce the amount of PD-L1 available for trans-binding interactions (cell to cell) with PD-1 expressed on T cells. This not only leads to diminished PD-1 inhibitory signaling [99] but also reduced CTLA4-CD80 immune regulatory axes [100]. However, the PD-L1/CD80 interaction does not obstruct the binding of CD80 to CD28. Consequently, they can form a trimeric complex that engages CD28 and thereby conveys a normal co-stimulatory signal [99] (Figure 3B). 

Exploring the availability of PD-L1/L2 to PD-1 in SLE could yield valuable insights, potentially positioning this interaction as a promising therapeutic target during autoimmune responses, as demonstrated in a recent study that the interaction between astrocytic PD-L1 and microglial PD-1 is necessary for the attenuation of autoimmune central nervous system inflammation in the acute and progressive stages of a mouse model of multiple sclerosis [101]. Also, T cells from the synovial fluid of patients with active rheumatoid arthritis exhibited decreased PD-1-mediated inhibition of their proliferation, especially at suboptimal concentrations of PD-L1. Treating those cells with PDL-1.Fc antibodies improved the reductions in their T-cell responses and ameliorated collagen-induced arthritis (CIA) [102]. In line with this, it was demonstrated that the disruption of PD-L1/CD80 cis binding by CD80 antibodies can alleviate autoimmunity due to the accessibility of the PD-1/PD-L1 interaction [103].

The inhibitory PD-1/PD-L1/2 pathway can also be restricted by the soluble form of PD-1 (sPD-1). sPD-1 is generated from an alternative splicing event, where exon 3, which contains the coding sequence for the transmembrane domain of PD-1, is deleted from the mRNA transcript. The expression of this transcript is upregulated following T-cell activation, leading to the release of sPD-1 into the plasma [104]. It has been reported that sPD-1 can interfere with PD-L1 and PD-L2, preventing the binding of these ligands to the transmembrane form of PD-1 and consequently blocking the negative signal [105] (Figure 3C). In this regard, elevated levels of sPD-1 have been found in patients with autoimmune diseases, such as SLE [106,107,108], RA [109,110,111], pSS [112,113], and the antineutrophil cytoplasmic antibody (ANCA)-associated vasculitides (AAV) [114]. This could promote a lack of cellular inhibition by blocking the inhibitory effect of membrane-bound PD-1 upon T-cell activation.

PD-L1 and PD-L2 can also be found in soluble forms (sPD-L1 and sPD-L2, respectively), which are generated through proteolytic cleavage of membrane-bound proteins by metalloproteases like ADAMs or metalloproteinases (MMPs) [115] or by alternative splicing that induces a protein product that lacks the transmembrane domain and is secreted in a soluble form [116,117]. In the context of cancer, it has been reported that sPD-L1 influences the activation of the inhibitory function of PD-1 expressed in T cells by binding to PD-1, leading to the activation of the downstream pathway [118]. However, the role of sPD-L2 in the activation of the inhibitory pathway remains unclear. Understanding the mechanisms of how sPD-L1 and sPD-L2 influence the pathogenesis of SLE and other autoimmune disorders is currently challenging due to varying reports. While some authors indicated elevated levels of these ligands compared to healthy controls, others reported decreased levels, and in some autoimmune disorders, there is a lack of sufficient information. In Table 1, we summarize some of the findings related to PD-1 and its ligands as well as their soluble forms in SLE and other autoimmune diseases.

## 6. Regulation of PD-L1 and PD-L2 Expression and Its Association with the Autoimmune Response

It is well known that the major inducers of PD-L1 and PD-L2 expression are interferon (IFN) cytokines, including type I (IFNα and IFNβ) and type II (IFNγ) interferons. An IFN pathway mediated by JAK/STAT signaling promotes the activation and expression of interferon regulatory transcription factor 1 (IRF1), which induces the amplification of the PD-L1/2 genes by binding directly to the PD-L1 promoter [137]. After PD-L1 (and probably PD-L2) translation, the protein can be modified via post-translational modifications, such as glycosylation, acetylation, tyrosine phosphorylation, and mono-ubiquitination, upon epidermal growth factor (EGF) stimulation [138]. Knowledge about post-translational modifications to PD-L1 has primarily stemmed from studies focusing on cancer. However, there are currently no works evaluating the effects of these modifications in the context of autoimmunity. Therefore, it will be important to understand the regulatory mechanisms of both PD-L1 and PD-L2, as they could serve as potential therapeutic targets.

The PD-L1 protein can be stabilized via glycosylation at the N35, N192, N200, and N219 residues, increasing the half-life of the protein and regulating the PD-1/PD-L1 interaction [139]. PD-L1 has an extra glycosylation site at N64 [88]. Palmitoylation at C272 [140,141] and phosphorylation at Try112 [142] have also been implicated in stabilizing the protein and cell surface distributions in some cancer cells. In contrast, some modifications like ubiquitination mediated by the CDK4 cyclin and phosphorylation at S195 of PD-L1 induce its degradation [142]. 

Recent findings have also demonstrated the association of two proteins members of the chemokine-like factor (CKLF)-like MARVEL (CMTM) family with the stability of PD-L1 expression in cancer and in normal myeloid cells in both humans and mice [143,144]. CMTM6 is a type 3 transmembrane protein that binds to PD-L1, maintaining its expression on the cell surface, without modifying the constitutive IFN-induced mRNA expression. CMTM6 is also found in recycling endosomes with PD-L1, preventing its ubiquitination and degradation by lysosomes. Consequently, the depletion of CMTM6 leads to a reduction in PD-L1 expression on the cell surface [143,144]. CMTM4 is another member of this family. It shares 55% sequence similarity with CMTM6 but has received less attention in research, despite also demonstrating functions similar to CMTM6 in the regulation of PD-L1 [144]. 

CMTM6 and CMTM4 are just beginning to be explored in autoimmune diseases. Using a genetic association network model, Davis et al. conducted a resequencing analysis of exomes from SLE patients and healthy individuals and identified CMTM4 as a strong candidate for study in SLE, as it emerged as an implicated susceptibility gene [145]. This discovery suggests that CMTM4 may play a role in the pathogenesis of SLE, potentially contributing to our understanding of this disease’s genetic basis and offering new avenues for research into its mechanisms and therapeutic interventions.

Another study that analyzed the serum levels of CMTM6 in pSS found increased concentrations of this molecule in pSS patients compared with non-pSS patients and healthy controls and found upregulated expression of CMTM6 in the labial glands of the pSS patients. In that study, CMTM6 was also positively correlated with sPD-1 and sPD-L1 [113]. Recently, it was reported that lower PD-L1 expression in the monocytes of patients with AVV was associated with increased lysosomal degradation of PD-L1 because of the reduced expression of CMTM6 [146]. Further studies on CMTM6 and CMTM4 are needed to understand their effects on PD-L1 and PD-L2 in the autoimmune context (Figure 4).

## 7. Is It Possible That PD-1/PD-L1/2 Has Another Important Role beyond Inhibition?

The binding of PD-1 with its ligands initiates a cascade of signals that results in the inhibition of T-cell signaling. When PD-1 binds to PD-L1/2, the ITIM and ITSM domains are phosphorylated. This phosphorylation leads to the activation of phosphatase Src homology region domain-containing phosphatase 2 (SHP2). SHP2 then dephosphorylates the kinases ZAP70, PI3K, and Ras, resulting in the restriction of the downstream TCR signaling pathway [20,21]. PD-1 can also inhibit CD28 co-stimulatory signaling by limiting PI3K/AKT, resulting in a strong inhibition of T-cell activation [147]. In B cells, the PD-1-PD-L1/2 interaction also results in the phosphorylation of the tyrosine residues in the ITIM/ITSM domains, which recruits SHP1/SHP2. SHP1/SHP2 further dephosphorylates B-cell receptor (BCR) proximal signaling molecules that attenuate the activation of the downstream pathway, leading to a loss of B-cell function [148]. However, despite the well-defined signaling pathway downstream of PD-1, the following question remains: is it possible that PD-1 may exert other effects beyond the inhibition of the cellular response?

In this context, a phosphoproteomic analysis revealed that not all pathways downstream of PD-1 are inhibitory. Instead, PD-1 ligation resulted in increased phosphorylation at several tyrosine sites, some of which were associated with the activation of cellular functions [149]. This notion was further supported by studies in which it was observed that following PD-1 ligation, the transcriptional signatures of proliferating CD4^+^ T cells were enriched with genes associated with an activated state. Furthermore, they observed that the genes induced after PD-1 ligation significantly differed between PD-L1 and PD-L2 stimulation [150,151]. These observations are consistent with the idea that PD-1 may be involved in regulating specific cellular functions beyond its inhibitory role.

Given that PD-1 is a marker for GC-resident Tfh cells [152] and that cTfh and Tph cells express high levels of PD-1 on their cell surfaces, as discussed above, it could be interesting to assess whether this protein plays a role in the differentiation or function of these cells, which have mainly been implicated in the pathogenesis of SLE and other autoimmune disorders.

The idea that PD-1 is important in germinal center responses, potentially leading to antibody production, has been gaining acceptance over the years. In 2010, Good-Jaboson et al. [153] demonstrated that PD-1 regulated the survival of germinal center B cells and the development of long-lived plasma cells. They observed that PD-1 was upregulated in Tfh cells, while PD-L1 and PD-L2 were upregulated in germinal center B cells. In their study, they also found that mice deficient in PD-1, PD-L2, or both PD-L1 and PD-L2 exhibited reduced numbers of long-lived plasma cells [153]. However, how a deficiency in PD-1 or its ligands is related to impaired humoral responses remains a question. They proposed that the loss of PD-1 signaling resulted in an increase in Tfh cells but reduced synthesis of IL-4 and IL-21 mRNA by these cells. This, in turn, led to a deficient cooperative response, as it is well known that IL-21 is required for the optimal differentiation and maturation of long-lived plasma cells and antibody production [154]. Support for this notion was obtained using *Pdcd1^−/−^* mice. It was demonstrated that a deficiency of PD-1 led to increased expression of BCL6 in Tfh cells and increased numbers of these cells within the germinal center. However, the expression of interferon regulatory factor 4 (IRF4), which is essential to the production of IL-21, was diminished in these cells. Consequently, Tfh cells from *Pdcd1^−/−^* mice produced less IL-21 than cells from wild-type (WT) mice [155]. More recently, a study conducted in bone marrow chimeras, where both PD-L1-sufficient and -deficient B cells were evaluated, also demonstrated that interactions between PD-1 on Tfh cells and PD-L1 on B cells optimized the competitiveness and affinity maturation of B cells by controlling the positioning and functioning of Tfh cells [156]. The authors proposed that PD-1 has a bystander signaling mode for the recruitment of Tfh cells into a follicle. This function is carried out in synergy with ICOS, another member of the B7 family of co-stimulatory molecules that is important for Tfh cell development [157]. Furthermore, it was discussed that both ICOS and PD-1 are required for optimal IL-21 production by Tfh cells, which could be a possible explanation for why these cells express high levels of PD-1 on their surfaces. However, it has also been demonstrated that IL-21 is essential to the production of PD-1 [158], which suggests a positive feedback loop. 

Most of the information available about the functions performed by PD-1 and its ligands has been obtained from studies involving CD8^+^ T cells in cancer, where CD8^+^ T cells expressing PD-1 have been considered as an exhausted T-cell subpopulation [159]. However, it appears that the functions of PD-1 may vary depending on the context of the disease and the T-cell subpopulation. In this sense, Tfh, cTfh, and Tph cells are distinguished by their high expression of IL-21 [49,53,160], the cytokine through which they perform their primary helper functions, along with the expression of other cytokines during the humoral response. Therefore, it is suggested to evaluate whether these cells can be considered an exhausted T-cell subpopulation, and new studies are necessary to elucidate the role of PD-1 beyond inhibition in CD4^+^ T cells in the context of autoimmune responses.

## 8. PD-1 as a Possible Therapeutic Target in SLE

Proposals to develop biological agents targeting co-inhibitory pathways to treat patients with autoimmune diseases are not new [161]. Based on the discussion above, it is clear that a better understanding of the PD-1/PD-L1/L2 axes in SLE and other autoimmune diseases could lead to new therapeutic strategies that can exclusively target the PD-1 pathway or can be combined with other treatments. Two therapeutic approaches could be followed: facilitating PD-1’s accessibility to its ligands and administering PD-1 agonists. 

### 8.1. Accessibility between PD-1 and Its Ligands

Daisuke Sugiura et al. generated anti-CD80 antibodies that dissociate the cis-PD-L1-CD80 complex. This dissociation allows PD-L1 to interact with PD-1, leading to the activation and potentiation of the inhibition of T cells and resulting in the attenuation of autoimmune symptoms [103].

### 8.2. PD-1 Agonists

Previously, a study was conducted using a murine model to evaluate the impact of systemically activating PD-1 through the administration of PD-L1-Ig in SLE-prone mice. The results demonstrated that PD-L1-Ig activated the PD-1 pathway, leading to the suppression of Th17 cell generation; decreases in cytokine levels, including IFN-γ, IL-17, and IL-10; and reduced production of anti-dsDNA autoantibodies. These effects collectively contributed to the mitigation of kidney disease and extended the survival of these SLE-prone mice [162]. Using computational design, a small molecule named PD-MP1 was created to specifically bind to both human and murine PD-1 at the PD-L1 interface. The trimerization of PD-MP1 acts as a PD-1 agonist, significantly inhibiting the activation of murine T cells. This suggests that PD-MP1 has potential as a PD-1 agonist for the treatment of autoimmune diseases and inflammatory conditions [163]. Also, PD-1 agonist molecules called ImmTAAI have been developed. These molecules can bind to target cells and imitate PD-L1, leading to highly effective activation of PD-1 receptors on interacting T cells and, therefore, immune suppression [164]. Nanoparticles coated with dexamethasone have shown therapeutic efficacy in improving lupus nephritis (LN) in MRL/lpr mice. They specifically target CD4^+^ T cells, activating PD-1 and TIGIT (T-cell immunoglobulin and ITIM domain) signaling, leading to the inhibition of effector T cells and enhancing the immunosuppressive activities of Tregs [165]. More recently, natural autoantibodies that downregulate specific autoimmune disorders have been discovered. These autoantibodies are called regulatory rheumatoid factor (regRF). regRF specifically binds to PD-1 on activated T cells, and it has been reported that serum containing regRF reduces the number of PD-1^+^CD4^+^ lymphocytes. Therefore, regRF utilizes the PD-1 pathway to regulate activated CD4^+^ T lymphocytes [166].

Although the potential of activating the PD-1 co-inhibitory pathway to alleviate autoimmunity is still being explored, it appears to be a promising therapeutic approach. In fact, there are PD-1 agonists in phase I and II clinical trials, and researchers are collecting information to propose novel strategies such as combining low doses of IL-2 with PD-1 agonists [167]. This has the potential to revolutionize conventional strategies and bring us closer to the era of precision medicine.

## 9. Conclusions

PD-1 is a molecule whose function appears to largely depend on the environment and context in which it is found. Most studies on PD-1 and PD-L1/2 expression, regulation, and function have observed CD8^+^ T cells in the pathogenesis of cancer. However, this review emphasized the importance of carrying out new assays to elucidate the mechanisms controlling PD-1’s function on CD4^+^ T cells during the autoimmune response, as it has been suggested that PD-1 could play a role in the differentiation and function of various T-cell subpopulations, such as Tfh, cTfh, and Tph cells, which are associated with autoimmune disorders. This suggests that the regulation of this molecule is more complex. 

We highlighted some of the most recent findings about the PD-1 and PD-L1/L2 axis in systemic lupus erythematosus to demonstrate its inhibitory and non-inhibitory functions and its association with disease activity. We suggest that a deeper understanding of the PD-1-dependent interactions between B and T cells could have value in identifying new therapeutic targets for this disease. 

## Figures and Tables

**Figure 1 ijms-25-07726-f001:**
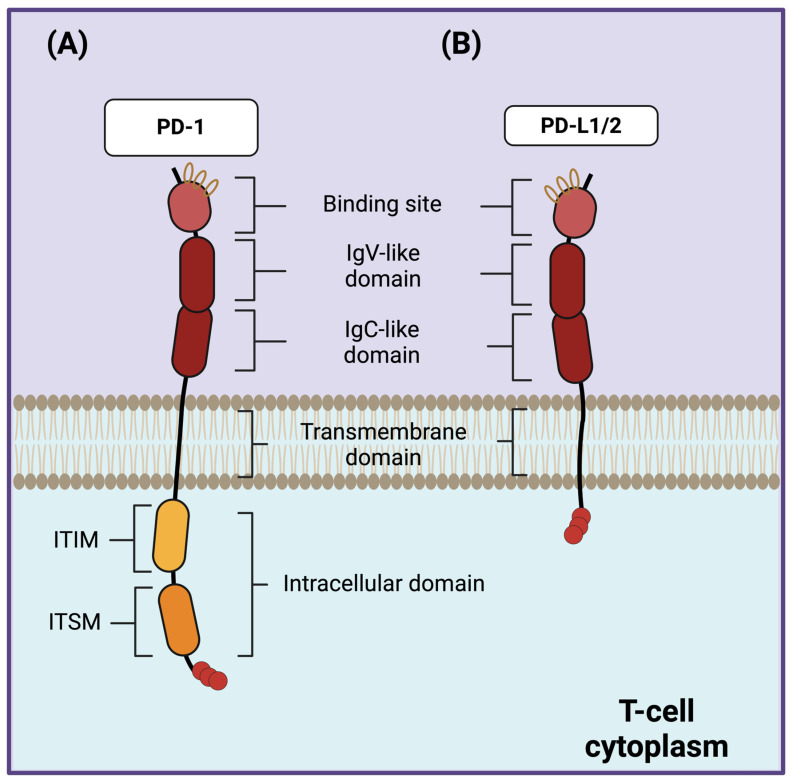
Structures of PD-1 and its ligands PD-L1 and PD-L2. (**A**) The structure of PD-1 is composed of IgV-like and IgC-like domains in the extracellular region, followed by a transmembrane domain, and the intracellular region is composed of ITIM and ITSM domains. (**B**) The structures of its ligands PD-1 and PD-L2 share their extracellular structure with PD-1. PD-1: programmed death 1; PD-L1: programmed cell death-ligand 1; PD-L2: programmed cell death-ligand 2; ITIM: immunoreceptor tyrosine-based inhibitory motif; ITSM: immunoreceptor tyrosine-based switch motif.

**Figure 2 ijms-25-07726-f002:**
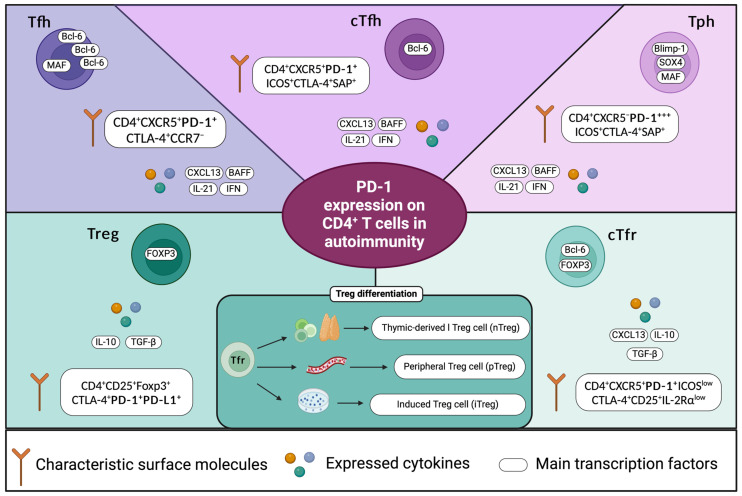
PD-1 expression on the surfaces of different CD4^+^ T cells in autoimmunity. PD-1 expression has been studied in different subpopulations of CD4^+^ T cells like Tfh, cTfh, Tph, Treg, and Tfr cells. Tfh: T follicular helper cells; cTfh: circulating T follicular helper cells; Tph: T peripheral helper cells; Treg: regulatory T cells; Tfr: follicular regulatory T cells; SAP: SLAM-associated protein.

**Figure 3 ijms-25-07726-f003:**
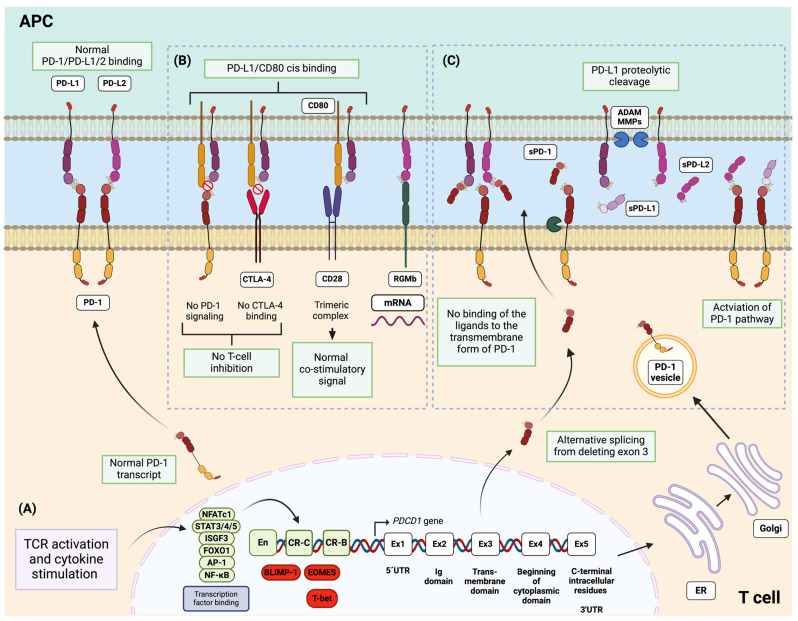
Expression of PD-1 and its ligands PD-L1 and PD-L2. (**A**) The expression of PD-1 is stimulated by various transcription factors generated after TCR activation and cytokine stimulation. (**B**) PD-L1 can also bind to CD80, and PD-L2 can also bind to RGMb. (**C**) PD-1 and its ligands can also be found in a soluble form generated by alternative splicing or proteolytic cleavage. PD-1: programmed death 1; PD-L1: programmed cell death-ligand 1; PD-L2: programmed cell death-ligand 2; TCR: T-cell receptor; RGMb: repulsive guidance molecule b.

**Figure 4 ijms-25-07726-f004:**
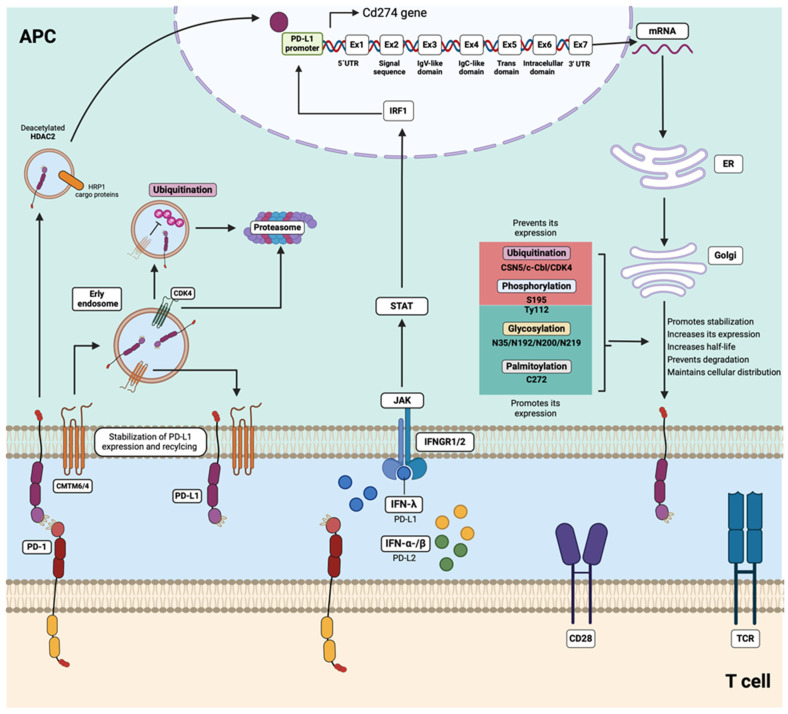
Regulation of PD-L1 expression. CMTM6/4 can bind to PD-L1 on the plasma membrane, maintaining its expression on the cell surface as well as in early endosomes. This interaction favors recycling, thereby preventing ubiquitination and degradation by proteasomes. In contrast, CDK4 promotes the ubiquitination of PD-L1. HDAC2 deacetylates PD-L1, facilitating its translocation to the nucleus, where PD-L1 can activate genes involved in the regulation of its own expression. Post-translational modifications, such as glycosylation (N35, N192, N200, and N219), phosphorylation (Try112), and palmitoylation (C272), stabilize PD-L1, increase its expression, extend its half-life, and maintenance its cellular distribution. Conversely, ubiquitination (CSN5, c-Cbl, and CDK4) and phosphorylation (S195) promote lysosomal and proteasomal degradation. HDAC2: histone deacetylase; CDK4: cyclin D/cyclin-dependent kinase 4; PD-1: programmed death 1; PD-L1: programmed cell death-ligand 1; CMTM6: CKLF-like MARVEL transmembrane domain-containing protein 6; IFN-y/a/β: interferon-gamma/alpha/beta; IRF1: interferon-gamma receptor; GAS: gamma interferon activation site; STAT: Signal Transducer and Activator of Transcription; JAK: Janus kinase; mRNA: messenger RNA; ER: endoplasmic reticulum.

**Table 1 ijms-25-07726-t001:** PD-1, PD-L1, and PD-L2 tissue, cell, and serum expression in systemic lupus erythematosus and other autoimmune diseases.

Disease	Evaluation	Finding	Reference
SLE	Tissue expression	PD-1 and PD-L1 were identified in the kidney of lupus nephritis patients.	Bertsias, George K et al., 2009 [119]
	Cell expression	Increased percentages of PD-1-expressing CD3^+^ T cells and PD-1-expressing CD19^+^ B cells in SLE. High expression of PD-L1-expressing CD19^+^ B cells and PD-L2-expressing CD14^+^ monocytes.	Liu, Ming-Fei et al., 2009 [120]
		Elevated frequency of PD-L1-expressing neutrophils in SLE. This percentage decreased after receiving a 15-day treatment with corticosteroids and immunosuppressive drugs.	Luo, Qing et al., 2016 [121]
		PD-L1 was significantly higher in CD19^+^ cells of SLE patients with active disease and LN. The expression of PD-L1 was increased in double-negative B cells (DN) and plasma cells (PC). The percentage of CD19^+^PD-L1^+^ cells was correlated with the disease activity index and the number of T follicular helper cells (Tfh).	Jia, Xiao-Yun et al., 2019 [122]
		Higher expression of PD-1, PD-L1, PD-L2, and CD86 in CD19^+^CD20^−^ B cells than in CD19^+^CD20^+^ B cells.	Zhu, Qingqing et al., 2021 [123]
		Higher expression of PD-1 and PD-L1 in CD11c^+^ B cells.	Rincon-Arevalo, Hector et al., 2021 [124]
	Soluble levels	Increased serum levels of anti-PD-1 IgG in new-onset SLE patients.	Shi, Hui et al., 2017 [125]
		Higher serum levels of sPD-1 and sPD-L1.	Du, Yan et al., 2020 [108]
		Higher serum levels of sPD-1 and sPD-L2 in SLE patients with high disease activity. sPD-L1 was not elevated in SLE patients.	Hirahara, Shinya et al., 2020 [107]
		Higher expression of membrane-bound PD-L2 on monocytes. Lower serum levels of sPD-L2.	Tong, Min et al., 2020 [126]
RA	Tissue expression	PD-L2 was highly expressed on macrophages in synovial tissue.	Xiong, Jian et al., 2023 [127]
		Low PD-L1 expression in RA synovial tissue.	Guo, Yanxia et al., 2018 [128]
		PD-1 expression on synovium-infiltrating lymphocytes. PD-L1 expression on cells lining synovium.	Matsuda, Kotaro et al., 2018 [129]
	Cell expression	Increased percentages of PD-L2-expressing CD14^+^ monocytes.	Xiong, Jian et al., 2023 [127]
		Increased PD-L1 expression on mDCs in synovial fluid compared with mDCs in peripheral blood.	Moret, Frederique M et al., 2014 [130]
	Soluble levels	Lower sPD-L2 in the serum of RA patients.	Xiong, Jian et al., 2023 [127]
		sPD-1 levels were increased in ACPA^+^ but not ACPA^−^ early RA.	Guo, Yanxia et al., 2018 [128]
		Elevated sPD-1 and sPD-L1 levels in serum and synovial fluid of RA patients.	Wan, Bing et al., 2006 [131]
		Higher sPD-1 expression in early and chronic RA.	Greisen, S R et al., 2014 [110]
		Elevated sPD-1 expression in serum of RA patients.	Bommarito, D et al., 2017 [132]
pSS	Tissue expression	PD-1 expression on infiltrating lymphocytes in salivary glands from 52% of SS patients. PD-L1 expression on ductal and acinar epithelial cells from 68% of SS patients.	Kobayashi, Masaya et al., 2005 [133]
		PD-1 and PD-L1 expression in labial glands were higher than in non-pSS controls.	Qian, Sirui et al., 2022 [113]
	Cell expression	Increased expression of PD-L1 in CD11c^+^ B cells of pSS patients.	Rincon-Arevalo, Hector et al., 2021 [124]
	Serum levels	Lower serum levels of sPD-L2 in pSS patients.	Loureiro-Amigo, Jose et al., 2021 [134]
		Elevated serum levels of sPD-L2 in pSS patients.	Nishikawa, Ayumi et al., 2016 [135]
		Elevated serum levels of sPD-1 and sPD-L1 in pSS patients.	Qian, Sirui et al., 2022 [113]
MS	Tissue expression	Astrocytes in white matter lesions from MS patients upregulated PD-L1 in response to aryl hydrocarbon receptor and interferon signaling.	Linnerbauer, Mathias et al., 2023 [101]
	Cell expression	Frequencies of PD-L1-expressing CD19^+^ B cells and PD-L1^+^/IL-10+CD14^+^ monocytes were higher in stable multiple sclerosis patients compared to acute MS (AMS) patients.	Trabattoni, Daria et al., 2009 [136]
	Serum levels	No information.	

SLE: systemic lupus erythematosus (SLE); RA: rheumatoid arthritis; pSS: primary Sjögren’s syndrome; MS: multiple sclerosis.

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
