# Peer review of "Exploring the Role of PD-1 in the Autoimmune Response: Insights into Its Implication in Systemic Lupus Erythematosus"

_ijms, 2024, doi:10.3390/ijms25147726_

Round 1

Reviewer 1 Report

Comments and Suggestions for Authors

I considered the manuscript entitled “Exploring the Role of PD-1 in the Autoimmune Response: Insights Into its Implication in Systemic Lupus Erythematosus” by Nefertari Sagrero-Fabela, et al, which is intended to be published in IJMS journal.     

The manuscript has some potential. However, before the manuscript might be reviewed, it needs a profound rewording by a specialized English medical writer. It is not logical that the reviewer, or the readers in the future, had to investigate the meaning of what is being expressed.

Only reading the abstract and the Introduction:

Despite advances in our knowledge of systemic lupus erythematosus has increased over time, It should be better: Even though advances in our knowledge of systemic lupus erythematosus have increased over time

I do not understand: Research has ventured into investigate the potential impact of PD-1

We aim to provide potential explanations for why PD-1 may not effectively carry out its suppressive functions, It should be better: Our goal is to provide possible explanations for why PD-1 may not effectively perform its suppressive functions

Comments on the Quality of English Language

It has to be refined before reviewing. Similar as you ask for help for the methods, the statistcal analysis and other parts of the study, you have to ask for helping for writing to specalized team

Author Response

Thank you very much for taking the time to review our manuscript (ijms-3090691). We appreciate all the comments and suggestions made by the reviewers. Please find the detailed responses below and the corresponding revisions/corrections.

Comment I: “I considered the manuscript entitled “Exploring the Role of PD-1 in the Autoimmune Response: Insights Into its Implication in Systemic Lupus Erythematosus” by Nefertari Sagrero-Fabela, et al, which is intended to be published in IJMS journal.      

The manuscript has some potential. However, before the manuscript might be reviewed, it needs a profound rewording by a specialized English medical writer. It is not logical that the reviewer, or the readers in the future, had to investigate the meaning of what is being expressed.”

Response I: Thank you for your valuable feedback. Our manuscript was reviewed by a MDPI English Editor. We attach the English editing certificate (ID: 82524). Our goal is to ensure clarity and precision, making it easily comprehensible for both reviewers and future readers. Your input has been fundamental in helping us achieve this standard.

Comment II: “Only reading the abstract and the Introduction: Despite advances in our knowledge of systemic lupus erythematosus has increased over time, It should be better: Even though advances in our knowledge of systemic lupus erythematosus have increased over time.

Response II: According to the English editor’s suggestions we changed the paragraph as follows: “Despite advances in understanding systemic lupus erythematosus (SLE), many challenges remain in unraveling the precise mechanisms behind the disease's development and progression”.

Comment III: “I do not understand: Research has ventured into investigating the potential impact of PD-1”

Response III: According to the English editor’s suggestions the paragraph was changed to: “Research has investigated the potential impacts of PD-1”.

Comment IV: “We aim to provide potential explanations for why PD-1 may not effectively carry out its suppressivefunctions, It should be better: Our goal is to provide possible explanations for why PD-1 may not effectively perform its suppressive functions”

Response IV: Thank you for the comment, we changed the paragraph according to your suggestion.

In addition, all the English editor’s suggestions were included in this manuscript. We hope that with these improvements, the manuscript now meets the standards of quality in writing.

Thank you again for your guidance.

Kind regards,

Dr. Claudia Azucena Palafox Sánchez

Reviewer 2 Report

Comments and Suggestions for Authors

Despite advances in understanding systemic lupus erythematosus (SLE), many challenges remain in unraveling the precise mechanisms behind the disease's development and progression. Recent evidence has questioned the role of programmed cell death protein 1 (PD-1) in suppressing autoreactive CD4+ T cells during autoimmune responses. The scientific literature has explored the impact of PD-1 on various CD4+ T cell subpopulations, including T follicular helper (Tfh) cells, circulating Tfh (cTfh) cells, and T peripheral helper (Tph) cells, which all show significant PD-1 expression and are closely associated with several autoimmune disorders, including SLE. This review highlights the complex role of PD1 in autoimmunity and underscores the need for further research to clarify its functions during autoreactive T-cell responses. Additionally, it discusses the potential of targeting PD-1 or its ligands as a therapeutic approach in SLE.

The review is well-written and comprehensive, with topics organized in a streamlined fashion. The authors have provided clear figures to help the reader better understand the text, along with a well-structured table. They have made a significant effort to summarize the existing literature on this topic. The manuscript is thorough and informative, making it a valuable contribution to the field. Given its current quality, the manuscript deserves publication in its current form.

Author Response

Thank you very much for taking the time to review our manuscript (ijms-3090691). We appreciate all the comments and suggestions made by the reviewers. Please find the detailed responses below and the corresponding revisions/corrections.

Comment I: Despite advances in understanding systemic lupus erythematosus (SLE), many challenges remain in unraveling the precise mechanisms behind the disease's development and progression. Recent evidence has questioned the role of programmed cell death protein 1 (PD-1) in suppressing autoreactive CD4+ T cells during autoimmune responses. The scientific literature has explored the impact of PD-1 on various CD4+ T cell subpopulations, including T follicular helper (Tfh) cells, circulating Tfh (cTfh) cells, and T peripheral helper (Tph) cells, which all show significant PD-1 expression and are closely associated with several autoimmune disorders, including SLE. This review highlights the complex role of PD1 in autoimmunity and underscores the need for further research to clarify its functions during autoreactive T-cell responses. Additionally, it discusses the potential of targeting PD-1 or its ligands as a therapeutic approach in SLE.

Response I: Thank you very much for taking the time to review this manuscript. We took your suggestions to improve the abstract's writing. Additionally, we submitted the document for editing and rewriting by an English editor by the MDPI system (ID: 82524). We attach the certificate to the English edition.

Comment II: The review is well-written and comprehensive, with topics organized in a streamlined fashion. The authors have provided clear figures to help the reader better understand the text, along with a well-structured table. They have made a significant effort to summarize the existing literature on this topic. The manuscript is thorough and informative, making it a valuable contribution to the field. Given its current quality, the manuscript deserves publication in its current form.

Response II: We are grateful for your recognition of the value of the information and organization presented, which we believe makes a significant contribution to the field. Your feedback is invaluable and motivates us to further enhance the quality of our work. Thank you once again for your comments.

Kind regards,

Dr. Claudia Azucena Palafox Sánchez

Round 2

Reviewer 1 Report

Comments and Suggestions for Authors

After english correction, the manuscript has greatly improved. I read the manuscrpt and greatly appreciate its content. It is ok por its publication.